# OpenReview forum: "Targeted Low-rank Refinement: Enhancing Sparse Neural Networks with Precision"
_ICLR.cc/2025/Conference — ICLR 2025 Conference Withdrawn Submission_

### Official Review · Reviewer_3RcV · 2024-10-30

**Soundness:** 3
**Presentation:** 3
**Contribution:** 2
**Rating:** 5
**Confidence:** 5

**Summary:**

This paper investigates methods to refine sparse LLMs without re-training. The authors propose using SVD to model the differences between the original large model and its sparse version with a low-rank matrix. They further develop an iterative updating strategy to optimize this low-rank matrix without altering the sparsity level of the model. Experiments on sparse models generated by techniques like magnitude and Wanda demonstrate the effectiveness of the proposed approach.

**Strengths:**

1. **Well-Written**: The paper is well-structured and written, presenting its methodology and proofs in a clear and rigorous manner.
2. **Efficient Methodology**: The proposed method is efficient, showing promising results in refining sparse models without re-training.

**Weaknesses:**

1. **Limited Experimental Scope:**
   - The experiments are primarily conducted on models up to 13 billion parameters. Larger models, such as those with 30 billion and 70 billion parameters, were not explored.
   - The method is only tested on sparse models generated by magnitude and Wanda. It lacks evaluation on sparse models processed by SparseGPT. Additionally, there is no downstream task results for Wanda, which limits a comprehensive understanding of its effectiveness.

2. **Lack of Comparative Analysis:**
   - The authors did not compare their method against similar types of method that refines sparse LLMs without re-training, such as DsNot[1], in terms of both time using and performance metrics.

3. **Methodological Concerns:**
   - While the proposed approach shows innovation, there is concern about whether it can outperform traditional fine-tuning methods like LoRA. It might also be beneficial to investigate if further fine-tuning the low-rank parameters, estimated by the proposed method, with LoRA can lead to even more exciting results.

[1] Dynamic Sparse No Training: Training-Free Fine-tuning for Sparse LLMs. In ICLR, 2024.

**Questions:**

Please see the weakness part.

---

### Official Review · Reviewer_j5M4 · 2024-11-01

**Soundness:** 3
**Presentation:** 3
**Contribution:** 1
**Rating:** 5
**Confidence:** 5

**Summary:**

This paper proposes a new method to improve pruned neural networks by using a low-rank approximation to capture essential information lost during pruning, bridging the performance gap between dense and sparse models. Unlike traditional pruning, this approach iteratively refines the sparse weight matrix with a low-rank adjustment, enhancing model accuracy, especially at high sparsity levels.

**Strengths:**

+ The proposed low-rankness plus sparsity is reasonable for compressing LLMs since we cannot train large models.
+ The low-rank part and sparse part can reduce the approximation error between the compressed model and the original model, thereby improving the compressed accuracy.
+ The paper provides multiple baselines and approximation strategies, as well as theoretical proofs.

**Weaknesses:**

- The novelty is not sufficient for this venue. The combination of low-rankness and sparsity is an old topic that has been explored for many years [R1, R2]. Applying the well-established approximation techniques to decompose/compress the large matrices in LLMs has little technical contribution. Besides, compressing DNN models using low-rank and sparse decomposition has already been well explored in [R3]. This paper just scales it to larger models and matrices.
- The accuracy comparison with respect to sparsity level is not fair. With the additional low-rank part, it is obvious that the accuracy would get improved. However, the authors do not show the storage cost of additional low-rank part.
- Except for 4:8 and 2:4 sparsity, other random sparsity is not meaningful, which would not get practical acceleration and will be even slower in the GPU execution.
- The authors have referred to [R4-R6]. I have not seen significant differences among them.
- The theoretical error bound is not tight and cannot guarantee anything in the practical assessment.


[R1] Sparse and Low-Rank Matrix Decompositions, Forty-Seventh Annual Allerton Conference, 2009.

[R2] Godec: Randomized low-rank & sparse matrix decomposition in noisy case. ICML 2011.

[R3] On compressing deep models by low rank and sparse decomposition, CVPR 2017.

[R4] Slope: Double-pruned sparse plus lazy low-rank adapter pretraining of llms

[R5] LoSparse: Structured Compression of Large Language Models based on Low-Rank and Sparse Approximation

[R6] OATS: Outlier-Aware Pruning Through Sparse and Low Rank Decomposition

**Questions:**

Please refer to the Weakness. Additionally, what are the practical model sizes in the experimental results? What are the practical speedups?

---

### Official Review · Reviewer_JsWA · 2024-11-02

**Soundness:** 2
**Presentation:** 2
**Contribution:** 2
**Rating:** 3
**Confidence:** 5

**Summary:**

The paper presents Targeted Low-Rank Refinement for enhancing the performance of sparse neural networks. By augmenting pruned models with low-rank adjustments, it bridges the gap between dense and sparse networks. They provide theoretical analysis and experiments on LLAMA to verify the efficacy of the approach.

**Strengths:**

- The idea of introducing low-rank adaptor to estimate the deviation between dense and sparse model is interesting.

- Writing is generally good and easy to follow.

**Weaknesses:**

- Experiments are not satisfactory. At first, the title is targeting on sparse neural network, while the experiments only target on LLMs. The author needs to add more CNN experiments to make the scope properly aligned. Secondly, they only report perplexity of LLM, which is not sufficient. They need to report more standard benchmark metrics for cross validation.

- Motivation of using low-rank matrices is not clear. In particular, why the difference between sparse and dense blocks are low-rank?

**Questions:**

See the weakness.

---

### Official Review · Reviewer_gtzD · 2024-11-04

**Soundness:** 3
**Presentation:** 3
**Contribution:** 1
**Rating:** 5
**Confidence:** 4

**Summary:**

This paper presents an approach to approximate a dense full matrix by combining a sparse matrix with a low-rank approximation of the residual error. It introduces an iterative algorithm that progressively refines the sparse weight matrix while incorporating a low-rank approximation, backed by theoretical analysis to demonstrate the convergence and effectiveness of the method.

**Strengths:**

1): The theoretical analysis is solid and contributes a valuable foundation for understanding the method’s validity.

2): The presentation is clear and well-organized.

**Weaknesses:**

1): The technical novelty is limited, as similar decompositions for mitigating compression error during fine-tuning have been explored in large language models, including methods like LQ-LoRA  [A1], LoQT[A2], LoftQ [A3], and etc.

2): The method operates in the weight space. It would be interesting to explore decomposition in the gradient space instead, similar to techniques like LoRAPrune [A4] and GaLore [A5].

3): The experimental results need improvement, particularly in demonstrating practical significance—a common issue in post-training structured pruning methods. For example, with a 13B model at 2:4 sparsity, the average accuracy drops from 40.4 to 31.2, which limits practical impact.

4): There are no comparisons with prior extensive structured pruning literature.


Reference:

[A1]: LQ-LORA: LOW-RANK PLUS QUANTIZED MATRIX DECOMPOSITION FOR EFFICIENT LANGUAGE MODEL FINETUNING, ICLR 2024

[A2]: LoQT: Low Rank Adapters for Quantized Training, ICML 2024 workshop

[A3]: Loftq: Lora-fine-tuning-aware quantization for large language models, ICLR 2024

[a4]: LoRAPrune: Structured Pruning Meets Low-Rank Parameter-Efficient Fine-Tuning, ACL 2024 findings

[A5]: GaLore: Memory-Efficient LLM Training by Gradient Low-Rank Projection, ICML 2024

**Questions:**

See weakness.

---

### Note · Authors · 2024-12-03

I have read and agree with the venue's withdrawal policy on behalf of myself and my co-authors.